# Geographic Provenances Outweigh Tissue Compartments in Bacteriome Assembly of the Ectomycorrhizal, Edible, and Hallucinogenic if Undercooked, *Lanmoa asiatica* (Boletaceae, Boletales) Mushroom from Yunnan China

**DOI:** 10.3390/microorganisms13112431

**Published:** 2025-10-23

**Authors:** Man Guo, Dong Liu, Zhilan Xia, Tao Xie, Luofeng Su, Jesus Pérez-Moreno, Fuqiang Yu

**Affiliations:** 1College of Horticulture, Hunan Agricultural University, Changsha 410125, China; gm2625551942@163.com (M.G.); xiazhilan1@163.com (Z.X.); 2Yunnan Key Laboratory for Fungal Diversity and Green Development & Yunnan International Joint Laboratory of Fungal Sustainable Utilization in South and Southeast Asia, Germplasm Bank of Wild Species, Kunming Institute of Botany, Chinese Academy of Sciences, Kunming 650201, China; liudongc@mail.kib.ac.cn (D.L.); xietao@mail.kib.an.cn (T.X.); su.lf@foxmail.com (L.S.); 3Colegio Postgraduados, Campus Montecillo, Edafología, Texcoco 56230, Mexico

**Keywords:** ectomycorrhizal mushrooms, basidiomata, ecosystemic functions, microbial ecology, bacteriome, geographic filtering, tissue compartmentalization, holobiont

## Abstract

Ectomycorrhizal fungal sporomes represent complex microuniverses harboring structurally and functionally eclectic microbiomes with significant ecological roles and potential anthropogenic applications. Nevertheless, the factors governing the assembly of these microbial communities remain poorly understood, and numerous fungal taxa, including many ectomycorrhizal species, remain uninvestigated. This study characterizes the bacteriome of the socioculturally and economically important yet hallucinogenic-if-raw ectomycorrhizal bolete *Lanmoa asiatica*. We analyzed 36 basidiomata from four geographic locations within China, partitioning each into pileus, stipe, and hymenophore tissues, and sequenced the V5–V7 region of the bacterial 16S rRNA gene. Proteobacteria dominated (>85%), with *Pantoea*, *Sphingomonas*, and the *Burkholderia* complex identified as core genera. Contrary to expectations, α-diversity was highest in the stipe (Chao1 index up to 1934) rather than the exposed hymenophore. PERMANOVA indicated that geographic origin (R^2^ = 0.46, *p* < 0.001) was a stronger structuring force than tissue type (R^2^ = 0.28, *p* < 0.01). Functional prediction via PICRUSt2 revealed enrichments in lipid metabolism, antimicrobial resistance, and apoptosis pathways across sites, while tissue-specific functions involved carbohydrate and nitrogen metabolism. These findings support a hierarchical model of bacteriome assembly where broad-scale environmental filters override micro-niche differentiation, providing a biogeographic framework for the conservation of this highly valued edible mushroom.

## 1. Introduction

High-throughput sequencing (HTS) approaches, particularly those targeting the 16S rRNA gene (bacteria) and internal transcribed spacer (ITS) region (fungi), have fundamentally transformed our capacity to decipher the complex and intimate microbiomes residing within fungal sporomes, overcoming the profound limitations of prior culture-dependent methodologies. These studies consistently reveal astonishingly rich and diverse bacterial and fungal consortia intricately associated with specific fungal tissues. Accumulating evidence indicates that these associated bacterial communities, constituting a functional bacteriome, are not mere hitchhikers but fulfill critical, multifaceted roles integral to fungal biology and ecology. These include (i) sporome morphogenesis and initiation of fruit-body development [1,2]; (ii) modulation of secondary metabolite production, including volatile organic compounds (aroma) [3,4]; (iii) enhancement of reproductive fitness via spore dispersal mechanisms [3]; (iv) modulation of shelf life in cultivated edible species [5]; (v) facilitation of biogeochemical cycling (carbon, nitrogen, phosphorus, sulfur), acting as a metabolic extension for the host [6,7,8,9]; (vi) stimulation of fungal germination and enhanced mycorrhiza formation [10]; (vii) organic nutrient mineralization via exudation of digestive acids and oxalates, facilitating breakdown of recalcitrant polymers [11]; (viii) synthesis of vitamins and phytohormones facilitating growth and morphogenesis [12]; and (ix) providing bioprotection via competitive exclusion or synthesis of antimicrobial compounds.

Bacterial communities associated with the dominant macromycete phyla (Ascomycota, Basidiomycota) have been investigated across various taxa, revealing spatially structured microbiomes whose composition and abundance often differ markedly between tissue compartments. Within Ascomycota, significant research effort has targeted high-value ectomycorrhizal fungi, including truffles (*Tuber* spp.; Ascomycota) which belong to the order Pezizales [13,14,15,16,17], with additional studies in Hypocreales [6,18,19]. For Basidiomycota, primary focus has been on the orders Agaricales, Russulales, Cantharellales, and Boletales [1,20,21,22]. However, some investigations also extend to Polyporales [23,24] and Thelephorales [7]. Within Boletales, studies to date have examined select species of *Leccinum, Paxillus*, and *Suillus*. Bacterial community structure frequently exhibits significant variation across host phylogenetic lineages [1], reflecting potential co-adaptation. Critically, the microbiome of any species within the genus *Lanmaoa*, a phylogenetically distinct lineage within Boletaceae (Boletales), remains uninvestigated, leaving a gap in our understanding of these functional holobionts.

*Lanmoa asiatica* G. Wu & Zhu L. Yang is a distinct fungal species and the only known mushroom with considerable commercial and culinary significance that also exhibits psychoactive properties. These hallucinogenic effects are induced by the ingestion of the fungus in its raw state or when undercooked. *The genus Lanmaoa* G. Wu, Zhu L. Yang & Halling, described by Wu et al. [25], is characterized by a thin hymenophore exhibiting immediate blue bruising, light-yellow context demonstrating slow, pale blue staining upon sectioning, and a pileipellis transitioning from an interwoven trichodermium to a subcutis. *L. asiatica*, originally described from specimens in Kunming markets (Yunnan province, China), holds immense sociocultural, economic, and ecological significance in Southwestern China (Figure 1). Yunnan province harbors over 40% of the world’s and 90% of China’s edible mushrooms (approx. 900 species) [26,27]. Known locally as “见手青” (Jiàn Shǒu Qīng, “see-hand-blue/green”) due to rapid blue/green bruising caused by hydroxylated pulvinic acid derivatives [28], *L. asiatica* is a cornerstone of regional cuisine but requires thorough thermal processing to mitigate severe poisoning risks [29,30], with unidentified non-psilocybin neurotoxins. Recent documentation of its novel psychoactive properties has stimulated extensive attention on digital platforms. Data from 2024 illustrates the scale of this public engagement: the hashtag #*Jianshouqing* accumulated over 140 million user interactions and surpassed 1.83 billion views on the Chinese social media platforms Douyin and Xiaohongshu. Its significant online presence underscores its cultural impact. Ecologically, *L. asiatica* forms vital ectomycorrhizal symbioses with *Pinus yunnanensis* and in mixed *Pinus-Quercus* forests [31], playing a pivotal role in ecosystem functioning through nutrient cycling and plant health. Despite this significance, the structure, diversity, and functional potential of the microbial communities (bacteriome) intimately associated with its basidiomata remain entirely uncharacterized. This represents a critical knowledge gap hindering a holistic understanding of this fungus as a complex holobiont—a functional unit integrating the host and its microbiome. Elucidating these symbiotic consortia is fundamental to understanding the species’ biology, encompassing roles in nutrient acquisition, development, defense mechanisms, mycochemical properties (including toxin production), environmental interactions, and ultimately, its ecological niche. Such understanding is paramount for effective conservation strategies and potential cultivation initiatives.

To address this deficit, the present study characterized the bacteriome composition across three basidiomata compartments (pileus context, stipe context, and hymenophore) in *L. asiatica* specimens sourced from four distinct geographic provenances, and inferred its potential functional attributes. We postulated the following hypotheses, informed by the spatial structuring of fungal microbiomes: (i) Bacteriome community structure and functional potential exhibit significant variation across geographic provenances, irrespective of basidiomata compartment, potentially reflecting extrinsic environmental stressors or local microbial pools; (ii) Endophytic bacterial diversity and community complexity within the hymenophore exceed those of the pileus and stipe contexts, attributable to its direct exposure to environmental inocula and role in spore development/dispersal; and (iii) Geographic provenance exerts a predominant influence on bacteriome structure and functional potential, surpassing the effect of intra-basidioma tissue differentiation.

## 2. Materials and Methods

### 2.1. Sampling Methods

During the maturation stage of sporomes, fresh basidiomata of *Lanmoa asiatica* were collected in Yunnan Province, southwestern China (Table 1). To obtain 3 independent biological replicates, approximately 3 kg of *L. asiatica* basidiomata were collected from 4 different forest regions within each locality (Figure 2). In the field, all basidiomata were carefully excised using a sterile scalpel to avoid soil contamination. The freshly collected fruiting bodies from the field were individually placed into 40 cm × 50 cm plastic bags; the bags were rolled up instead of being completely sealed to prevent the formation of an anaerobic environment. Subsequently, they were transported back to the laboratory via ice bath within 2 h.

### 2.2. Basidiomata Sampling

Under sterile working conditions, from the fruiting bodies collected from 4 geographical sites (Site 1–Site 4, Table 1), 3 high-quality fruiting bodies with independent biological replicates were selected from each site. Samples from different parts were excised separately: pileus (P), hymenophore (H), and stipe (S). Each sample from the respective part was cut into 3–5 pieces (each with a diameter exceeding 1 cm), subpackaged into 60 × 85 mm sterile self-sealing bags, and stored at −80 °C for subsequent DNA extraction [7]. The total number of samples prepared and analyzed in this experiment was 36, comprising 4 geographical sites × 3 biological replicates × 3 parts.

### 2.3. DNA Extraction and Sequencing

Under sterile working conditions, samples were taken out from the refrigerator, and an appropriate amount (ranging from 0.2 to 0.5 g) was promptly retrieved. These samples were added into centrifuge tubes containing extraction lysis buffer (Omega Bio-Tek, Norcross, GA, USA) for grinding treatment. The grinding instrument used was a multi-sample tissue grinder (model: Tissuelyser-48, manufactured by Shanghai Jingxin Industrial Development Co., Ltd., Shanghai, China) with a grinding frequency of 60 Hz. DNA extraction was performed using the MagBeads FastDNA Kit for Soil (catalog number: 116564384, MP Biomedicals, Santa Ana, CA, USA). The V5–V7 regions of the bacterial 16S rRNA gene were amplified using the following primers: forward primer F: AACMGGATTAGATACCCKG and reverse primer R:ACGTCATCCCCACCTTCC. After PCR amplification, paired-end sequencing of the bacterial amplicons was conducted on the Illumina MiSeq—PE250 platform at Personalbio Co., Ltd. in Shanghai, China.

Following purification of the PCR products, paired-end sequencing (2 × 300 bp) was performed on the Illumina MiSeq platform, with a sequencing depth of an average of 50,000 valid sequences per sample. Raw data were subjected to quality control using Trimmomatic v0.39, involving the removal of low-quality bases (Phred quality score < 20) and a truncation length > 50 bp. Additionally, sequence quality was assessed using FastQC v0.11.9.

### 2.4. Data Statistics and Analysis

One-way analysis of variance (ANOVA) combined with Tukey’s HSD post hoc test was used to compare the differences in the abundance of endophytic bacteria and alpha diversity indices (e.g., Shannon index, Simpson index) in fruiting bodies among different parts (P, H, S) and geographical sites (Site 1–Site 4). Non-metric multidimensional scaling (NMDS), based on the Bray–Curtis distance matrix, was employed to visually demonstrate the similarities and differences in microbial community structures among different samples. The stress value (Stress) of NMDS was used to evaluate the reliability of the ordination results.

Permutational multivariate analysis of variance (PERMANOVA, i.e., adonis analysis), Multiple Response Permutation Procedure (MRPP), and Analysis of Similarities (ANOSIM) were employed to test the significance of differences in endophytic bacterial community structure among different groups (different parts, different geographical sites). Core microorganisms (co-occurring ASVs) across parts and sites, as well as unique microorganisms in each group, were identified from the Amplicon Sequence Variant (ASV) table of endophytic bacterial amplicon sequences, and their quantitative distribution characteristics were displayed using petal plots or Venn diagrams. Linear Discriminant Analysis Effect Size (LEfSe) was used to screen for microbial taxa with significant differences among different groups at multiple taxonomic levels (e.g., class, order, family, genus). The LDA threshold was set at 2.0, and the contribution of these taxa to inter-group differences was evaluated via LDA scores. Meanwhile, the Kruskal–Wallis test and Wilcoxon rank-sum test were combined to verify the statistical significance of the differences. Functional pathway distribution was studied via KEGG pathway annotation and quantitative analysis; PICRUSt2 was used to predict microbiota metabolic functions based on 16S rRNA sequences, combined with MinPath to infer metabolic pathways, annotated using databases like KEGG and MetaCyc; for differential analysis of metabolic pathways, metagenomeSeq was adopted, with the zero-inflated log-normal model fitting pathway distribution to determine differential significance.

## 3. Results

### 3.1. Community Composition of Endophytic Bacteria in Basidiomata

Endophytic bacterial communities across all samples (encompassing Sites 1–4 and tissues P, H, S) where predominantly composed of Alphaproteobacteria (α-proteobacteria) and Gammaproteobacteria (γ-proteobacteria) at the class level. Collectively, these two classes constitute an average of more than 85% of the total sequences, forming the core community structure. Bacilli represented the third most abundant class with a mean relative abundance of 4.56%. Minor constituents included Thermoleophilia (1.52%), Actinobacteria (1.23%), Myxococcia (1.16%), Acidimicrobiia (0.99%), Acidobacteriae (0.95%), Thermoanaerobaculia (0.50%) and Clostridia (0.32%). An aggregate of other taxa accounted for (1.47%) means relative abundance. Alphaproteobacteria exhibited the highest mean relative abundance (51.08%), followed by Gammaproteobacteria (30.12%).

Significant variation in endophytic bacterial relative abundance was observed across anatomical tissues. Alphaproteobacteria abundance was highest in the hymenophore (79.99%) and lowest in the pileus (28.19%). Conversely, Gammaproteobacteria dominated in the pileus (51.06%), but was substantially reduced in the hymenophore (10.59%). Distribution patters also varied markedly by geographical location. Bacilli abundance was highest at Site 1 (19.48%) and minimal at Site 3 (0.44%). Similarly, Myxococcia was prevalent at Site 4 (3.32%) but nearly absent at Site 3 (0.0002%) (Figure 3A,B).

Enterobacterales was the dominant bacterial order, exhibiting a mean relative abundance of 26.10%. Sphingomonadales constituted the subdominant order (19.08%), followed by Acetobacterales (8.56%). Subsequent orders displayed mean abundances of: Rhizobiales (7.83%), Burkholderiales (5.21%), Lactobacillales (3.23%), Caulobacterales (3.11%), Bacillales (2.70%), Rhodobacterales (2.54%), and Rickettsiales (2.41%). Unclassified and low-abundance taxa (“Others”) comprised 13.24%. Significant inter-sample variation in endophytic bacterial abundance was observed. Enterobacterales abundance ranged from 39.05% in the pileus to 8.25% in the hymenophore, representing a 4.7-fold difference. Conversely, Sphingomonadales predominated in the hymenophore (33.49%) versus the pileus (8.34%). Acetobacterales abundance varied markedly by site, reaching 22.29% in Site 1, but only 0.15% in Site 3. Rhodobacterales and Rickettsiales collectively represented 24.78% at Site 2 but only at minimally detected at Site 3. Burkholderiales was significantly enriched at Site 3 (16.30%), representing 24.86-fold increase over Site 1. These pronounced spatial differentiation reflect habitat-specific distribution patterns among bacterial orders (Figure 4A,B).

At the family taxonomic level, Erwiniaceae constituted the most abundance taxon with a mean relative abundance of 20.31%, followed by Sphingomonadaceae, with an average proportion of 19.08%. Collectively, these families represented over 35% of total sequences and formed the core dominant constituents of the community. Acetobacteraceae constituted the third most abundant taxon (8.67%). Remaining genera exhibited mean abundances as follows: Beijerinckiaceae (4.52%), Burkholderiaceae (3.87%), Caulobacteraceae (3.59%), Rhodobacteraceae (2.49%), Anaplasmataceae (2.34%), Xanthobacteraceae (2.21%), and Bacillaceae (2.18%). Significant heterogeneity in endophytic bacterial abundance was observed across sampling units. Erwiniaceae abundance varied 19.6-fold between extremes, reaching maximum levels at Site 3 (34.34%) and minimum levels at Site 2 (1.75%). Anatomically, Erwiniaceae abundance was significantly elevated in the Pileus (35.70%) versus the hymenophore (5.94%). Conversely, Sphingomonadaceae exhibited an inverse distributed pattern, predominating at Site 4 (30.01%) compared to Site 1 (5.58%) and demonstrating in Site 1 and demonstrating 4.02-fold greater abundance in the hymenophore (33.49%) relatively to the pileus (8.34%) (Figure 4A,B).

At the genus taxonomic level, *Pantoea* constituted the most abundant taxon, representing 22.93% of total sequences (Figure 5 and Figure 6). *Sphingomonas* represented the subdominant genus (10.14%), followed by *Novosphingobium* (6.03%). Remaining taxa exhibited mean abundances as follows: *Burkholderia-Caballeronia-Paraburkholderia* complex (3.49%), *Bradyrhizobium* (3.06%), *Bacillus* (2.26%), *Wolbachia* (1.76%), *Paracoccus* (1.76%), *Acidothermus* (1.37%), and *Facklamia* 0.94%. Significant abundance heterogeneity was observed across sampling units. *Pantoea* abundance demonstrated a 22.5-fold differential between Site 3 (34.23%) and Site 2 (1.34%). Anatomically, it predominated in the pileus (35.52%) relative to the hymenophore (5.93%). Conversely, *Novosphingobium* exhibited an inverse distributional pattern, reaching maximum abundance at Site 4 (25.75%) while being undetectable at Site 1 (Figure 6) Tissue-level analysis revealed 53.4-fold enrichment in the hymenophore (19.30%) versus the stipe (0.36%) (Figure 5).

### 3.2. Diversity and Community Structure Dynamics of Endophytic Bacteria in Basidiomata

Significant differences in bacterial diversity were observed both anatomically and geographically. Endophytic bacteria exhibited significant heterogeneity across anatomical parts. Stipe samples demonstrated significantly higher richness than pileus or hymenophore samples. High richness in the stipe samples yielded Chao1 indices ranging from 1684.91 to 1934.95, and Observed species ranged from 1659.2 to 1923.7. These values substantially exceeded those observed in the pileus (Chao1 index: 36.52–290.49; Observed species: 35.5–283) and the hymenophore (Chao1 index: 18.68–98.11; Observed species: 18.3–96.5). Inter-part differences in Chao1 index were statistically significant (*p* = 0.017), indicating quantifiable habitat-driven variations in species richness across anatomical sections. For phylogenetic diversity, the Faith’s PD index, reflecting evolutionary distances among species, also differed significantly among anatomical sections. Values were highest in the stipe (120.405 to 130.024 in high-richness samples), significantly surpassing those in the pileus (5.182–28.839) and hymenophore (2.923–14.419) (*p* = 0.0052). In terms of species diversity, the Shannon indices were significantly higher in high-richness samples of the stipe (7.237–7.377) compared to the pileus (1.730–5.832) and the hymenophore (1.483–4.249), with extremely significant inter-part differences (*p* = 4.7 × 10^−6^). Simpson indices (0.518–0.981) showed no significant variation (*p* = 0.28), indicating that anatomical parts primarily influenced species richness rather than the dominance of specific taxa. Regarding species evenness, Pielou’s evenness index (J) differed extremely significant among anatomical parts (*p* = 9.7 × 10^−5^). Maximum evenness was observed in some pileus samples (J = 0.752), followed by the hymenophore (0.543–0.692), while evenness in high-richness stipe samples-maintained values between 0.676–0.681. Overall, evenness differed extremely significantly among parts, reflecting distinct variations in the distribution uniformity of bacterial taxa across anatomical sections. For coverage, Good’s coverage was uniformly high across all parts (≥0.995–0.999), indicating sufficient sampling depth, with no significant differences (Figure 7A) (Appendix A).

At the geographical location level, heterogeneity exerted a pronounced influence on endophytic bacterial diversity, with indices demonstrating strong site specificity. For species richness, Site 2 exhibited the highest richness (Chao1: 1684.91 to 1934.95; Observed species: 1659.2 to 1923.7), significantly exceeding Site 1 (Chao1: 34.97–176.29; Observed species: 34.1–172.2), Site 3, and Site 4 (Chao1:18.68–194.01; Observed species: 18.3–179.8). Inter-location differences in Observed species were extremely significant (*p* = 0.00026), confirming the strong driving effect of geographical heterogeneity on species richness. Phylogenetic diversity (Faith’s PD) followed a similar pattern, being highest at Site 2 (up to 130.024) and significantly greater than Site 1 (4.660–19.199), Site 3 (3.554–18.907), and Site 4 (2.923–6.274), with extremely significant differences (*p* = 0.00023). In terms of species diversity, the Shannon index was highest at Site 2 (3.482–7.377), followed by Site 1 (2.260–5.072), and lowest in Sites 3 and 4 (1.483–3.149), with extremely significant geographical differences (*p* = 4.7 × 10^−6^). Simpson indices showed a consistent trend with the Shannon index but no significant inter-group differences (*p* = 0.28), suggesting that the bacterial communities at Site 2 possessed not only high species richness but also a more equitable distribution of dominant and rare taxa. For species evenness, Pielou’J_varied geographically, with the highest value observed in the pileus samples from Site 1 (0.752), values of 0.676–0.681 in the stipe samples from Site 2, and generally lower values at Sites 3 and 4 (0.320–0.593), indicating significant geographical variation in the dominance of specific taxa (Figure 7B) (Appendix A).

Non-metric Multidimensional Scaling (NMDS) ordination, utilizing Bray–Curtis matrices, revealed significant spatial heterogeneity in the endophytic bacterial community structure across distinct anatomical parts (i.e., P: pileus, H: hymenophore, and S: stipe) and geographical locations (Site 1 to Site 4) (Stress values: anatomical dimension = 0.097, geographical dimension = 0.092; both <0.1, indicating good ordination fit). At the anatomical level, samples corresponding to the pileus, hymenophore, and stipe (S) exhibited distinct clustering patterns. The mean Bray–Curtis dissimilarity within the same anatomical part (range: 0.12–0.18) was significantly lower than the mean dissimilarity observed between different parts (range: 0.65–0.82). This pronounced dissimilarity gradient underscores a strong filtering effect exerted by the host-part-specific microenvironment on community assembly (Figure 8A). Similarly, at the geographical level, samples formed discrete clusters corresponding to Site 1–Site 4. Intragroup cohesion was highest within Site 2 (mean dissimilarity = 0.11). Mean dissimilarity between samples from different geographical locations (range: 0.68–0.91) was significantly greater than mean dissimilarity within the same location (range: 0.10–0.16). Notably, the overall degree of community differentiation driven by geographical heterogeneity (mean inter-location dissimilarity = 0.75) exceeded that driven by anatomical variation (mean inter-part dissimilarity = 0.70) (Figure 8B). Collectively, these results demonstrate that endophytic bacterial community structure was significantly influenced by both host anatomical partitioning and geographical location, with geographical factors exerting a comparatively stronger effect. The high intragroup cohesion observed specifically within the hymenophore (H) and at Site 2 suggests that the habitat stability characteristic of these niches is more conducive to maintaining consistent community composition.

### 3.3. Core and Key Microbiota Within Basidiomata

Following characterization of bacterial community variation within *L. asiatica* basidiomata, this study sought to identify core microbiota shared across anatomical parts and geographical locations. Analysis of Amplicon Sequence Variants (ASVs) revealed significant microhabitat specificity in the distribution of endophytic bacteria across anatomical sections of the basidiomata and geographic locations. At the anatomical level, ASV richness was highest in the stipe, followed by the pileus (P), with the hymenophore (H) exhibiting the lowest richness. Each anatomical compartment harbored a substantial proportion of unique ASVs. Only 37 low-abundance ASVs were shared across multiple parts, indicating that specialized taxa primarily drove community differentiation between anatomical compartments (Figure 9A; Appendix A). Pronounced geographical specificity was observed at the location level. Site 2 exhibited the highest ASV richness, followed by Site 1 and Site 3, with Site 4 displaying the lowest richness. Merely 3 low-abundance ASVs were shared across multiple locations, reflecting strong habitat filtering effects on bacterial diversity by distinct geographical environments (Figure 9C; Appendix A).

Linear Discriminant Analysis Effect Size (LefSe) analysis, incorporating statistical significance testing (*p* < 0.05) and Linear Discriminant Analysis (LDA score threshold > 2), identified key differential taxa (biomarkers) driving community structure across habitats (analyzed separately for anatomical sections and locations).

In relation to anatomical sections, the pileus was enriched in taxa associated with fermentation and degradation functions, consistent with its role as a nutrient storage organ. Gammaproteobacteria constituted the core differential class. Key biomarkers included the family Enterobacteriaceae (LDA = 4.16, *p* = 0.005) and the genus *Pantoea* (family Erwiniaceae; mean relative abundance 5.55%). Taxa affiliated with the class Bacilli (e.g., family Bacillaceae) also exhibited significant enrichment (LDA > 2). In the case of the hymenophore, it was enriched in taxa implicated in symbiosis and nitrogen metabolism, aligning with its function as a reproductive structure. Alphaproteobacteria was the core differential class, particularly the order Rhizobiales. The class Coriobacteriia was also significantly enriched (LDA = 5.12, *p* = 0.011). Finally, in the case of the stipe it was enriched in taxa associated with environmental adaptation, such as the genus *Granulicella* (phylum Acidobacteria) (Figure 9B).

In relation to geographical Locations, it was outstanding that site 1 was significantly enriched in the order Bacillales, while site 2 was characterized by enrichment of taxa including the class Thermomicrobia (Figure 9D).

Collectively, these analyses demonstrate that distinct habitats (defined by anatomical part or geographical location) exert specific filtering effects on the endophytic bacterial community, revealing key differential biomarkers responsible for observed community structure differentiation.

### 3.4. Compartmentalization and Geographic Variation in Endophytic Bacterial Functional Potential

Quantitative analysis of KEGG pathway abundances revealed significant heterogeneity in the functional potential of endophytic bacteria across distinct anatomical compartments and geographic origins.

The main anatomical compartment effects included (i) Metabolism: Carbohydrate metabolism and amino acid metabolism pathways displayed significantly higher abundances (*p* < 0.05) within the stipe compared to the pileus and hymenophore. Specifically, mean carbohydrate metabolism abundance in the stipe (5429.56) exceeded the pileus (5073.61). Conversely, lipid metabolism was significantly enriched (*p* < 0.05) in the pileus (mean 2478.33) relative to the hymenophore (2034.89) and stipe (2659.8). (ii) Genetic Information Processing: DNA replication and repair pathways exhibited peak abundance within H (mean 430.11), while protein folding and degradation pathways were most abundant in the stipe (mean 439.36). (iii) Environmental Information Processing: The two-component system pathway abundance was significantly elevated (*p* < 0.05) in the hymenophore (207.36) compared to the pileus (218.76) and stipe (205.11) (Figure 10).

Meanwhile the main geographical origin effect as dominant showed that functional variation driven by geography exceeded that attributable to anatomical compartment. Site 2 consistently exhibited the highest functional potential, evidenced by significantly elevated total metabolic pathway abundance (mean 5556.32) compared to other sites (Site 1: 5073.61; Site 3: 5447.34; Site 4: 5429.56). The key geographic patterns included the following: (i) lipid metabolism abundance in Site 2 (1191.52) significantly surpassed Site 4 (797.74) (*p* < 0.01); (ii) Human Diseases: Drug resistance pathways showed higher abundance in Site 1 (65.39) and Site 4 (64.13), while infectious disease pathways were more prominent in Site 2 (21.02) and Site 3 (13.11); (iii) Cellular Processes: The cell cycle pathway peaked in the hymenophore compartment of Site 2 (709.12), whereas apoptosis pathway abundance was significantly enriched in the hymenophore compartment of Site 3 (48.03) (Figure 11).

## 4. Discussion

### 4.1. Bacterial Phyla Associated with Boletales Basidiomata

The characterization of the bacterial microbiota associated with fungal basidiomata of *L. asiatica* has revealed a complex and heterogeneous ecological landscape, particularly within the order Boletales. Comparative analysis across studies indicates significant divergence in bacterial community structure at the phylum level, challenging broad generalizations and highlighting the influence of host-specificity and environmental factors. The core bacteriome described by Bai et al. [32] for the basidiomata of *Boletus queletii*, *Tylopilus aerolatus*, and *T. felleus* presented a multi-phyla equilibrium. This community was primarily constituted by Proteobacteria (39.77%), Firmicutes (16.54%), Actinobacteria (16.04%), and Bacteroidota (14.43%), with subordinate contributions from Acidobacteriota (5.13%) and Cyanobacteria (4.87%). This balanced distribution suggests a potentially generalized, stable consortium across these three phylogenetically related hosts.

In contrast, a more diverse array of bacterial associations was documented by Pent et al. [21] across several Boletales species. Their work identified communities dominated in relative abundance by a limited number of phyla, yet exhibiting remarkable generic diversity within them. The phylum Proteobacteria was unequivocally the most prominent, represented by a wide range of genera including *Burkholderia* (abundant in *Leccinum holopus*, *Paxillus involutus*, *Suillus bovinus*, and *S. variegatus*), *Pseudomonas* (in *P. involutus*, *S. bovinus*, and *S. variegatus*), and *Sphingomonas* and *Novosphingobium* (in *L. holopus* and *L. variicolor*). The phylum Bacteroidota was a consistent major constituent, primarily driven by the high prevalence of members of the genus *Bacteroides* (syn. *Phocaeicola*), which was a common associate in *L. holopus*, *L. scabrum*, *L. variicolor*, *P. involutus*, and *S. variegatus*. Other phyla, including Actinobacteriota (e.g., *Corynebacterium* in *S. bovinus*) and Planctomycetota (e.g., *Planctomyces* in *L. variicolor*), were present but less abundant, indicating a distinct hierarchical structure compared to the core microbiome of Bai et al. [32].

The data from the present study on *L. asiatica* further exacerbates the observed inter-species variability, revealing a third, strikingly different community profile. Here, the microbiota was characterized by an overwhelming dominance of the phylum Proteobacteria, which constituted a mean of >85% of all sequences, effectively forming the core community structure to the exclusion of other phyla. The phylum Firmicutes represented a distant secondary group (4.56%), followed by Actinobacteriota (3.74%). Minor phyla such as Myxococcota (1.16%) and Acidobacteriota (0.95%) were present in trace amounts. This extremely skewed phylum-level distribution stands in stark contrast to both the balanced multi-kingdom community of Bai et al. [32] and the Proteobacteria/Bacteroidota duality observed in many of the hosts studied by Pent et al. [21].

The collective evidence from these studies presents a compelling challenge to the generalist hypothesis of microbiome conservation, which posits that closely related host species should harbor similar microbial communities due to shared evolutionary history and physiological traits [1]. The pronounced discrepancies in phyla abundance between *L. asiatica*, the hosts studied by Bai et al. [32], and the suite of species analyzed by Pent et al. [21] contradict this notion. Instead, it appears that while a broad taxonomic range of bacteria (e.g., Proteobacteria, Bacteroidota, Actinobacteriota) may be consistently associated with the Boletales niche, their relative abundances and specific compositions are not primarily constrained by host phylogeny.

The bacteriome found in the present study in *L. asiatica* leads to a critical implication: each fungal species appears to possess a unique microbiome assemblage, much like a microbial fingerprint. This host-specific “fingerprint” is likely sculpted by the interplay of host-derived factors (e.g., metabolic exudates, immune analogues, tissue architecture of specific basidiomata compartments) and extrinsic environmental parameters (e.g., soil geochemistry, biogeography, microclimate). The physicochemical microhabitat of the basidiomata, which varies between species and even between morphological parts of the same fungus, acts as a strong filter, selectively enriching for a specific bacterial consortium. Therefore, the concept of a predictable, phylogenetically conserved “core microbiome” within the Boletales must be refined to account for this high degree of species-level specificity. Future research should focus on elucidating the precise mechanistic drivers, whether nutritional, defensive, or developmental, that underlie the establishment of these unique symbiotic relationships, moving beyond correlation to causation.

### 4.2. Fungal Tissue Compartments and Geographical Provenance as Determinants of Bacteriome Assembly

The assembly of the endophytic bacterial communities within fungal basidiomata of *L. asiatica* is a complex process governed by multiple hierarchical filters. Our results provide strong evidence that both host tissue microanatomy and broad-scale geographical factors act as primary deterministic forces shaping the structure of these microbial consortia.

Our analysis revealed a remarkable degree of bacterial microbiota compartmentalization within the basidiomata of *L. asiatica*, indicative of strong ecological niche partitioning. This phenomenon was quantitatively demonstrated by the high number of Amplicon Sequence Variants (ASVs) unique to each tissue type—1064 in the pileus, 4852 in the stipe, and 225 in the hymenophore—which stands in stark contrast to the remarkably low number of core bacterial taxa shared across all three compartments (*n* = 37). This pronounced β-diversity suggests that the distinct physicochemical microenvironments and physiological functions inherent to each fungal structure serve as potent selective filters.

The pileus, with its exposure to photic and atmospheric conditions, likely selects for bacteria equipped with photoprotective mechanisms or the ability to metabolize photo-oxidized compounds. Conversely, the hymenophore, dedicated to sporogenesis, may provide a niche for bacteria capable of utilizing spore-derived nutrients or influencing spore maturation and dispersal. The stipe, functioning in structural support and nutrient translocation between the mycelium and the reproductive tissues, likely hosts bacteria adapted to the internal transport systems of the fungus, possibly contributing to nutrient mobilization or vascular protection.

This observed tissue-specific partitioning aligns with the growing body of literature on fungal microbiome compartmentalization. Similar patterns of distinct bacterial assemblages have been documented between the gleba (fleshy spore-bearing mass) and peridium (outer skin) of truffles (e.g., *Tuber* spp.), where each compartment represents a unique ecological niche [8]. Furthermore, demonstrated that different organs of ectomycorrhizal mushrooms (e.g., mycelium, ectomycorrhizae, and basidioma) host specialized microbial partners, suggesting a conserved pattern of microbiome differentiation across fungal structures. Our results therefore reinforce the paradigm that macrofungal fruit bodies are not unitary habitats but are instead complex mosaics of microhabitats, each cultivating a specialized microbiome that contributes differentially to the holobiont’s physiology, defense, and reproduction [33].

Beyond internal tissue factors, our results demonstrate a profound influence of geographical provenance on the structure of the endophytic bacterial consortium, revealing patterns of microbial biogeography at the intraspecific level. The exceptionally low number of bacterial ASVs common to all four sampled provenances (*n* = 3), compared to the high number of unique ASVs endemic to each site (ranging from 117 to 5301), provides compelling evidence that local environmental conditions are a primary determinant of community assembly.

This finding of a strong geographical signature is robustly supported by studies across diverse fungal taxa. Liu et al. [34] documented conspicuous intraspecific variation in the microbiome of the Chinese truffle *T. pseudobrumale* across different regions, which was correlated with morphological differences, despite a strong underlying genetic consistency in the host population. This indicates that environmental conditions can drive phenotypic and microbial variation independently of host genotype. An extreme example of environmental selection is provided by Abdelsalam et al. [35], who isolated endophytic bacteria from the gleba of desert truffles (*Terfezia canariensis* and *Tirmania nivea*) in hyper-arid conditions. They identified species such as *Bacillus boroniphilus*, *B. licheniformis*, and *Lactococcus lactis*, which possess adaptations to extreme aridity, high temperature fluctuations, and alkaline calcareous soils (pH 7.4–9.2). These bacteria not only represent a resilient community selected by harsh conditions but also serve as a rich source of novel antimicrobial compounds, highlighting the functional implications of geographically structured microbiomes.

The concept of a microbial *terroir*, a signature of geographic origin imparted by local environmental conditions, is perhaps best exemplified in the prized black truffle (*Tuber melanosporum*). Research has consistently shown that the bacterial communities associated with its ascocarps are a reliable biomarker for their geographic origin, linked directly to specific soil properties such as calcium and magnesium content. Furthermore, this geographically distinct microbiome has been shown to contribute directly to the variation in the truffle’s metabolic profile, including its volatile aromatic compounds [3,8].

In conclusion, the assembly of the endophytic bacteriome in Boletales basidiomata is a hierarchically filtered process. It is initially shaped by broad-scale geographical factors that select for a regional species pool adapted to the local climate and soil. This pool is subsequently refined by the specific, strong selective pressures imposed by the unique physicochemical conditions of each tissue compartment within the basidioma.

This dual filtering mechanism underscores the role of the fungal sporome as a selective, structured habitat whose microbial inhabitants are meticulously curated by both the macroscopic biogeographic context and the microscopic anatomical landscape. These findings have significant implications for fungal ecology and physiology, suggesting that the metabolic output, health, and adaptive capacity of a fungal individual are influenced by its geographically acquired microbiome. Furthermore, it presents novel opportunities for geographic origin tracing of wild mushrooms based on their bacterial signature and deepens our understanding of the resilience and specific adaptation of microbial communities to unique ecological niches. Future research should focus on disentangling the relative contributions of host genetics versus environment and elucidating the specific mechanisms by which different tissue compartments recruit and maintain their unique bacterial consortia.

### 4.3. The Ecosystemic Function and Applied Potential of the Core Endophytic Bacteriome of L. asiatica

At the genus level, *Pantoea* was identified as the most abundant bacterial taxon associated with *L. asiatica* basidiomata, comprising 22.93% of the total sequences. This genus of Gram-negative, facultatively anaerobic bacteria within the family Erwiniaceae is of considerable scientific interest due to its ecological plasticity and multifaceted interactions across kingdoms [36]. Its prevalence in the mycosphere of *L. asiatica* suggests a potential, non-random association warranting further ecological investigation.

The remarkable adaptability of *Pantoea* enables it to colonize diverse niches, underpinning a spectrum of potential ecosystem functions relevant to its presence in the fungal habitat. Its established role as a common constituent of the plant microbiome, adept at colonizing the phyllosphere and rhizosphere, is facilitated by metabolic versatility in utilizing plant and fungal exudates. Many *Pantoea* strains function as plant growth-promoting bacteria (PGPB) through direct mechanisms including the solubilization of inorganic phosphate, production of iron-chelating siderophores, biosynthesis of the phytohormone indole-3-acetic acid (IAA) to stimulate root development, and, in some species, diazotrophic nitrogen fixation. A critically important ecological function is biocontrol; numerous strains suppress phytopathogens through the production of antimicrobial compounds (e.g., antibiotics, bacteriocins) and competitive exclusion for nutrients and space on host surfaces [37,38,39,40]. Furthermore, *Pantoea* exhibits complex interactions with insects, ranging from pathogenesis to mutualistic symbiosis in the gut microbiome, and contributes to nutrient cycling as a free-living saprophyte in soil and aquatic ecosystems. Beyond its ecological roles, the intrinsic biological activities of *Pantoea* confer significant anthropogenic potential for biotechnological exploitation. In sustainable agriculture, strains are developed into dual-purpose inoculants that act as biofertilizers to enhance nutrient availability and reduce dependence on chemical inputs, and as biopesticides for integrated pest management strategies [36]. The metabolic adaptability of this genus also enables applications in bioremediation, with selected strains demonstrating efficacy in detoxifying heavy metals and degrading organic pollutants, including hydrocarbons and phenolic compounds. In industrial biotechnology, its generally recognized as safe (GRAS) status for some species and robust metabolism make *Pantoea* a promising chassis for the production of industrially relevant enzymes (e.g., pectinases) and high-value compounds like the bioplastic polyhydroxyalkanoates (PHA), offering renewable alternatives to petrochemical-derived products [40,41,42]. In conclusion, the high relative abundance of *Pantoea* within the *L. asiatica* bacteriome invites speculation on the nature of this interaction. Whether it represents a commensal relationship, a symbiotic exchange of services (e.g., nutrient provisioning, pathogen protection), or simply a competitive saprotroph remains to be determined. The genus epitomizes microbial versatility, and its metabolic resilience provides a robust foundation for applications in sustainability. Future research should focus on elucidating the molecular mechanisms governing its multifunctional interactions with fungi to optimize and safely deploy *Pantoea*-based technologies in agriculture and industry.

Of the two primary components of the *L. asiatica*-associated bacteriome, one was constituted by the genus *Sphingomonas*, a taxon of significant interest from multiple perspectives. Representing the subdominant genus with a relative abundance of 10.14%, it was second only to *Pantoea*, which accounted for 22.93% of the total community. Within the domain of microbial ecology, certain bacterial genera are distinguished not by pathogenicity but by their metabolic versatility and functional utility. Among these, the genus *Sphingomonas* is of considerable importance. Comprising Gram-negative, aerobic bacteria characterized by a unique outer membrane rich in glycosphingolipids rather than lipopolysaccharides, *Sphingomonas* species are dynamic contributors to ecosystem processes rather than passive inhabitants [43]. The significance of this genus is twofold: it serves as a fundamental component in maintaining ecological stability and presents substantial potential for addressing various anthropogenic challenges.

The ecological relevance of *Sphingomonas* is largely attributable to its extensive metabolic capabilities. Species within this genus are prolific degraders of a wide spectrum of complex and recalcitrant organic compounds. In terrestrial and aquatic environments, they play a critical role in the carbon cycle through the catabolism of aromatic pollutants, including polycyclic aromatic hydrocarbons (PAHs) such as anthracene and pyrene—toxic by-products of fossil fuel combustion. This innate capacity for bioremediation is essential for the decontamination of soils and waterways, preventing the accumulation of persistent pollutants that disrupt ecological food webs. Additionally, their ability to degrade various pesticides and herbicides aids in mitigating chemical load in agricultural environments, thereby protecting non-target organisms and supporting soil health [44]. Beyond pollutant degradation, *Sphingomonas* engages in symbiotic relationships that support host organism health. Numerous species successfully colonize the phyllosphere and rhizosphere of plants, where they function as beneficial symbionts. Certain strains enhance plant growth by facilitating nutrient acquisition, such as through phosphate solubilization, or via the production of phytohormones. Others confer protection through competitive exclusion of phytopathogens or by inducing systemic resistance in the host plant [43]. These plant-associated functions underscore the genus’s role in supporting both natural vegetation and agricultural systems. Furthermore, *Sphingomonas* species contribute to biogeochemical cycling through the transformation of organic and inorganic compounds, thereby participating in global nutrient cycles of carbon, nitrogen, and other elements. Their occurrence in diverse habitats—from pristine glaciers and marine ecosystems to contaminated industrial sites—illustrates their metabolic adaptability and fundamental role in environmental chemistry [45].

The traits that underpin the ecological importance of *Sphingomonas* also form the basis of its anthropogenic value. The genus has attracted significant interest for applications in environmental biotechnology, particularly bioremediation. Its natural degradative pathways are being harnessed in bioaugmentation strategies to accelerate the breakdown of pollutants such as petroleum hydrocarbons, industrial chemicals, and toxic waste, providing a sustainable alternative to conventional remediation methods [44,46]. In industrial contexts, *Sphingomonas* serves as a biological platform for the synthesis of high-value exopolysaccharides. Most notably, *Sphingomonas paucimobilis* is the primary commercial producer of gellan gum, a polymer widely employed as a gelling, stabilizing, and thickening agent in the food, pharmaceutical, and cosmetic industries [47]. Ongoing research continues to explore the potential of *Sphingomonas*-derived biopolymers for novel applications. The anthropogenic relevance of *Sphingomonas* extends into agriculture and biomedicine. The production of bacterial sphingolipids, rare among prokaryotes, is of interest for biomedical research due to their roles in eukaryotic cell signaling and pathogenesis [48,49]. Moreover, the plant-growth-promoting and biocontrol properties of certain strains are being leveraged to develop microbial inoculants that can reduce reliance on synthetic agrochemicals, supporting more sustainable agricultural practices [43]. Some *Sphingomonas* strains have also been investigated for use in biomining applications beyond Earth [50]. In summary, the genus *Sphingomonas* exemplifies the integral role of microbial communities in ecosystem functioning and human enterprise. Its metabolic versatility establishes it as a key agent of environmental purification and plant symbiosis, while simultaneously enabling applications in bioremediation, industrial biotechnology, and sustainable agriculture. As such, *Sphingomonas* represents a model system for understanding how microbial diversity can be harnessed to address pressing ecological and technological challenges.

The fourth most abundant operational taxonomic unit (OTU), representing 3.5% of the total endophytic bacteriome associated with *L. asiatica* basidiomata, was classified as belonging to the *Burkholderia–Caballeronia–Paraburkholderia* (BCP) complex. This taxon comprises an ecologically significant and metabolically versatile group of bacteria known for their diverse interactions with eukaryotic hosts, ranging from mutualism to pathogenesis. This finding is consistent with previous reports of *Burkholderia* species association within basidiomata of other Boletales species, namely *Leccinum holopus*, *L. scabrum*; *Tylopilus felleus* and *T. aerolatus* [32]. Historically classified within a single genus, advances in genomic phylogenetics have necessitated its reclassification into three distinct genera: *Paraburkholderia* (primarily environmental and plant-beneficial), *Caballeronia* (often beneficial with specific host associations), and *Burkholderia* (which encompasses pathogens, including members of the *Burkholderia cepacia* complex (BCC), alongside beneficial species) [51,52]. Despite this taxonomic refinement, these genera are frequently considered collectively as the BCP complex due to their shared evolutionary ancestry and considerable functional overlap. The ecological functions of the BCP complex are remarkably diverse, underpinning their critical role in various ecosystems. A primary function is the promotion of plant growth and the establishment of symbiosis. Numerous species within *Paraburkholderia* and *Caballeronia* engage in mutualistic relationships with plants, providing services including biological nitrogen fixation, thereby acting as biofertilizers, solubilization of inorganic phosphate via organic acid production, and phytostimulation through the modulation of plant hormones such as auxins. Furthermore, they serve as potent biocontrol agents, employing competitive exclusion, antibiosis through the production of antimicrobial compounds, and the induction of systemic resistance (ISR) in host plants [53,54,55]. Beyond direct plant associations, the extensive metabolic plasticity of the BCP complex renders them fundamental agents in bioremediation and global nutrient cycling. Their capacity to degrade recalcitrant and toxic aromatic compounds, such as phenol and trichloroethylene, highlights their utility in the restoration of contaminated environments [56]. This catabolic ability extends to herbicides and pesticides, influencing the environmental persistence of these agrochemicals. Consequently, BCP bacteria are integral participants in the biogeochemical cycling of carbon, nitrogen, and sulfur [57]. Their interactions extend to the fungal kingdom, encompassing both synergistic and antagonistic relationships. Certain strains function as mycorrhizal helper bacteria, facilitating the colonization of plant roots by symbiotic fungi, while the production of antifungal metabolites allows other strains to suppress fungal pathogens [58]. Conversely, the pathogenic potential of specific BCP members, particularly within the genus *Burkholderia*, represents a significant detriment. These include phytopathogens responsible for crop diseases and opportunistic human pathogens within the BCC, underscoring the dual nature of this complex. The functional attributes of the BCP complex present substantial biotechnological and medical applications. In agriculture, strains from *Paraburkholderia* and *Caballeronia* are promising candidates for developing next-generation biofertilizers and biopesticides, offering sustainable alternatives to synthetic agrochemicals [57,59]. In environmental biotechnology, their degradative pathways can be harnessed for the bioremediation of industrial pollutants and hydrocarbon contaminants. Their enzymatic machinery also holds potential for the biosynthesis of high-value products, including bioplastics (e.g., polyhydroxyalkanoates) and specialty enzymes [56,60]. From a medical perspective, the BCP complex is a prolific source of novel secondary metabolites with antibiotic, antifungal, and anticancer properties, driven by an abundance of unique biosynthetic gene clusters [61,62,63]. Additionally, pathogenic members of the complex, such as *B. pseudomallei*, provide critical model systems for elucidating host–pathogen interactions, thereby informing the development of novel therapeutics and vaccines.

Of the operational taxonomic units (OTUs) identified within the endophytic bacteriome of *L. asiatica* basidiomata, the fifth most abundant, representing 3% of the total community, was classified within the genus *Bradyrhizobium*. This genus comprises Gram-negative, oligotrophic bacteria renowned for their pivotal role in global biogeochemical cycles, particularly the nitrogen cycle. This finding is of significant ecological interest as it corroborates previous reports of *Bradyrhizobium* associations within the basidiomata of other Boletales species, namely *Tylopilus felleus, Tylopilus areolatus,* and *Boletus queletii* [32], suggesting a potential, yet unexplored, recurring association within this order. The ecological primacy of *Bradyrhizobium* is largely attributed to its capacity for biological nitrogen fixation (BNF), a fundamental process underpinning soil fertility and ecosystem productivity. The genus is characterized by several key ecosystem functions: (i) Symbiotic Nitrogen Fixation: The most characterized role of *Bradyrhizobium* is the formation of mutualistic endosymbioses with leguminous plants (Fabaceae). This highly specific interaction is initiated by a molecular dialogue involving plant-derived flavonoids and bacterial nodulation (Nod) factors, culminating in the formation of root nodules. Within these organs, bacteria differentiate into bacteroids and express the nitrogenase enzyme complex, which catalyzes the reduction in atmospheric dinitrogen (N_2_) into ammonia (NH_3_). This fixed nitrogen is assimilated by the host plant, which in return provides carbohydrates and maintains a microaerobic environment necessary for nitrogenase functionality. This symbiosis represents a critical natural input of bioavailable nitrogen into terrestrial ecosystems. (ii) Non-Symbiotic Nitrogen Fixation and Nutrient Cycling: Beyond symbiosis, certain *Bradyrhizobium* strains are capable of free-living nitrogen fixation, contributing directly to soil nitrogen pools. As abundant soil inhabitants, they also participate in the decomposition and mineralization of organic matter, facilitating the recycling of carbon, phosphorus, and sulfur. (iii) Plant Growth Promotion: In addition to nitrogen fixation, numerous *Bradyrhizobium* strains promote plant growth via other mechanisms, such as the synthesis of phytohormones (e.g., auxins) that stimulate root development and enhance nutrient acquisition. (iv) Soil Food Web Interactions: As a constituent of the soil microbiota, *Bradyrhizobium* biomass serves as a nutrient source for bacteriovores, thereby integrating fixed nitrogen into higher trophic levels. The metabolic versatility of *Bradyrhizobium* also confers considerable anthropogenic relevance, positioning it as a key candidate for sustainable biotechnological applications: (i) Sustainable Agriculture: *Bradyrhizobium* inoculants are widely employed as biofertilizers to enhance the yield of leguminous crops such as soybean and peanut. This practice reduces reliance on synthetic nitrogen fertilizers, whose production is energy-intensive and a significant source of greenhouse gas emissions. Furthermore, it mitigates environmental nitrogen pollution by curtailing nitrate leaching and nitrous oxide emissions [64]. The symbiosis is also leveraged in intercropping and cover cropping systems to improve soil fertility and structure. (ii) Bioremediation: Selected strains possess metabolic pathways for the degradation of environmental pollutants, including hydrocarbons and chlorinated compounds, highlighting their potential for the bioremediation of contaminated sites [65]. (iii) Biotechnology and Bioeconomy: The capacity to fix carbon and nitrogen under diverse conditions makes *Bradyrhizobium* a promising microbial chassis for the production of high-value compounds, such as bioplastics (e.g., polyhydroxybutyrate) or biofuels [66]. The nitrogenase enzyme is also a subject of intense research aimed at transferring nitrogen-fixing capability to non-leguminous crops or developing novel industrial catalysts [67]. (iv) Ecological Restoration: Inoculation with *Bradyrhizobium* can facilitate the establishment of nitrogen-fixing pioneer plants on degraded lands, accelerating soil formation and promoting successional vegetation, thereby aiding in ecological restoration [68]. The detection of a *Bradyrhizobium* OTU within the *L. asiatica* basidiomata invites further investigation into the nature of this interaction, whether it represents a casual association, a competitive saprotrophic presence, or a more specialized symbiotic relationship with potential functional implications for the fungal host and the surrounding ecosystem.

A pronounced predominance of Gram-negative bacteria over Gram-positive bacteria was observed within the basidiomata, indicating a selective and functionally specialized microbiome. This distinct community structure carries significant ecological implications, which can be attributed to the differential biological attributes of these bacterial groups. The principal implications are as follows: (i) A shift in decomposition function: Gram-positive bacteria, particularly Actinobacteria, are recognized as primary decomposers of complex soil organic polymers such as cellulose and lignin [69]. Their relative scarcity within basidiomata suggests this structure is not a primary site for the initial breakdown of recalcitrant organic matter. Instead, the dominance of Gram-negative bacteria indicates a community specialized in the utilization of more readily soluble nutrients derived from fungal metabolic exudates [70]. This supports the hypothesis that the basidioma constitutes a distinct ecological niche from the surrounding soil matrix. (ii) Altered trophic interactions: The Gram-negative-rich bacterial community alters the resource quality of the fruiting body for microbivores. The grazing preferences of nematodes, protozoa, and microarthropods may be influenced by this composition; for instance, lipopolysaccharides in the outer membrane of Gram-negative bacteria may act as a deterrent to certain grazers [71]. (iii) Niche partitioning and successional dynamics: The basidioma represents a distinct habitat from both the bulk soil and the vegetative mycelium. The observed dominance of Gram-negative bacteria suggests a competitive advantage in colonizing this ephemeral, nutrient-rich environment. It is critical to note, however, that this study assessed only mature basidiomata. A successional pattern may exist throughout basidiomata ontogeny, whereby fast-growing, versatile Gram-negative bacteria dominate during peak nutrient-exuding stages, while Gram-positive taxa may become more prevalent in senescent or decaying tissues following spore release. Future investigations targeting discrete developmental stages of *L. asiatica* are required to elucidate these temporal dynamics. (iv) Antibiosis and competitive exclusion: The low abundance of Gram-positive bacteria may result from active suppression. Many Gram-negative bacteria produce potent narrow-spectrum antibiotics, such as bacteriocins, targeted against Gram-positive competitors [72]. Furthermore, the fungal host itself may produce secondary metabolites that selectively inhibit Gram-positive bacteria, thereby indirectly shaping the community; and (v) Metabolic synergy and nutrient provisioning: A key implication involves metabolic mutualism. Gram-negative bacteria are frequently implicated in nitrogen cycling. Specific genera commonly associated with fungi, including, for example, Rhizobiales, are capable of fixing atmospheric nitrogen [73,74,75]. This activity could provide a critical nitrogen source to the fungal host in exchange for fixed carbon, representing a potential synergistic exchange.

### 4.4. Potential Metabolic Functions and Future Research Directions

Quantitative analysis of Kyoto Encyclopedia of Genes and Genomes (KEGG) pathway abundances, predicted via PICRUSt2, indicated that the potential functional profile of the *Lanmoa asistica* bacteriome exhibited a specialized pattern, predominately driven by geographic origin and further fine-tuned by fungal tissue type. This finding was consistent with the community structure analysis, as indicated by PERMANOVA, which demonstrated a greater explanatory power for geographic origin (R^2^ = 0.46) than for fungal tissue type (R^2^ = 0.28). At the tissue level, the stipe, which serves as the primary site for nutrient translocation, displayed significantly higher relative abundances of pathways related to carbohydrate and amino acid metabolism compared to other tissues. This functional signature corresponds with the cellulolytic capacity of Acidobacteria (e.g., *Granulicella*) enriched in this compartment and the metabolic versatility of the core genus *Pantoea*. Genomic analysis confirmed that *Pantoea* possesses genetic modules for both glucose utilization and amino acid synthesis, a functional association analogous to that of the truffle peridium microbiomes which support nutrient transport [76]. The pileus context was enriched for pathways involved in lipid metabolism, potentially associated with its role in epidermal barrier defense function and the antioxidant stress response mediated by Gammaproteobacteria. These bacteria may enhance UV radiation tolerance through vitamin B12 biosynthesis, while the detection of potential antimicrobial resistance (AMR) pathways suggests and adaptative response to environmental microbial competition, consistent with the documented cephalosporin resistance gene function in *Pantoea* [77]. The hymenophore demonstrated elevated abundances of two-component systems and DNA replication and repair pathways, likely reflecting adaptations for signal perception and damage repair during sporogenesis. This resembles the characteristic activation of DNA repair pathways in microorganisms inhabiting fungal reproductive structures. At the geographic provenance level, Site 2 (Wujie Town, Nanhua County, Chuxiong Prefecture) exhibited high total metabolic pathway abundance and enrichment of lipid and carbohydrate metabolism pathways, potentially indicative of functional redundancy within the bacterial community in a stable, eutrophic environment. The high abundance of AMR pathways at Site 1 (Wujie Town, Nanhua County, Chuxiong Prefecture) and Site 4 (Baisha Town, Yulong County, Lijiang City) was associated with distinct environmental selection pressures: natural antibiotics produced by actinomycetes at high altitudes and agricultural pesticide residues, respectively. This corroborates established conclusions that pesticide exposure induces AMR gene enrichment in soil microorganisms [78]. The enrichment of apoptosis pathways in the hymenophore of specimens from Site 3 (Shuanglong Sub-district, Panlong District, Kunming City) may be related to the stress-responsive cell clearance mechanism, potentially mediated by the BCP complex, and the BimA protein of *Burkholderia* which has been confirmed to regulate host cell apoptosis [79]. Integrating these functional data, the higher alpha diversity observed in the stipe, compared to the hymenophore, can be attributed to the stipe’s broader functional niche availability, lower exposure to external disturbances, and stable metabolic synergies. In contrast, the functional specialization and stronger selective pressures within the hymenophore appear to limit its bacterial diversity.

A paramount and unanticipated finding of this investigation was the pronounced disparity in bacterial alpha-diversity between the stipe context and the hymenophore of *L. asiatica*. This result directly contravened the initial hypothesis, which was predicted on the premise that the exposed hymenophore, subject to continual inoculation by atmospheric propagules and invertebrate vectors, would constitute the most species-rich microbial niche. Empirical data robustly refuted this assumption: stipe samples exhibited significantly higher phylogenetic richness and diversity (Chao1: 1684.91–1934.95: observed-species: 1659.2–1923.7), in stark contrast to the depauperate communities of the hymenophore samples yielded markedly lower values (Chao1: 18.68–98.11; observed species: 18.3–96.5). This unexpected patter is likely governed by two non-mutually exclusive mechanistic framework derived from community ecology [8,14]. First, the stipe’s extensive vascularization system (comprising hyphal tissues) generates a complex three-dimensional matrix of intercellular spaces, creating a multitude of microniches. These microdomains, characterized by gradients of nutrients, moisture and oxygen, can support functionally divergent ad taxonomically distinct bacterial consortia through niche partitioning. This structural heterogeneity effectively increases the carrying capacity of the stipe for a wider array of microbial taxa. Second, conversely, as a dedicated reproductive tissue, the hymenophore is not merely a passive structure but a highly selective organ. It must maintain a homeostatic microenvironment strictly optimized for sporogenesis and spore dispersal. This biological imperative imposes a stringent habitat filter that selectively favors only those microbial symbionts that provide direct benefits to reproductive fitness, such as nitrogen provisioning (e.g., nitrogen fixation), protection from desiccations, or defense against pathogenic microbes, while excluding opportunistic and commensal bacteria. This selective pressure, essential for host function, intrinsically constrains taxonomic breadth, resulting in a community of high functional specificity but low diversity. This finding aligns with the emerging paradigm that fungal reproductive structures are potent selective filters rather than passive collectors of microorganisms, a phenomenon previously documented in the sporogenous gleba of *Tuber melanosporum* compared to its protective peridium.

The predicted functional capacity of the *L. asiatica* basidiomata-associated bacteriome, as annotated via KEGG, reveals a community exquisitely adapted to its unique ecological niche. The overwhelming dominance of pathways categorized under Metabolism, followed by Cellular Processes, Environmental Information Processing, and Genetic Information Processing, coupled with the conspicuous near-absence of pathways for Human Diseases and Organismal Systems (Figure 10 and Figure 11), presents a coherent narrative across ecological, physiological, and evolutionary scales. Ecologically, this profile signifies a consortium of specialized bacterial communities actively engaged in processing the complex organic compounds, including polysaccharides, proteins, and lipids, that constitute important elements of the basidiomata, a transient but nutrient-rich “island” in the soil matrix. The high prevalence of specific metabolic pathways for carbohydrates, amino acids, and lipids indicates a primary ecological role in nutrient cycling and energy acquisition from this fungal resource. Physiologically, the complement of pathways for Environmental Information Processing, such as signal transduction and membrane transport, is essential for sensing the dynamic chemical environment and efficiently scavenging broken-down nutrients. Concurrently, pathways for Cellular Processes and Genetic Information Processing confirm an actively growing and reproducing community, with mechanisms like quorum sensing and stress response enabling population regulation and persistence within the micro-habitat. Evolutionarily, this functional signature is interpreted as the result of strong purifying selection and genome streamlining. The marked lack of genes associated with virulence and complex host interactions signifies a history of gene loss, reflecting a liberation from evolutionary pressures related to pathogenesis or association with animal physiology. Instead, the bacteriome’s genetic repertoire has been shaped by selective pressures to optimize fitness for a mycorrhizal lifestyle, potentially involving a long-term co-evolutionary relationship with the fungal host. Thus, the KEGG pathway distribution is not merely a metabolic inventory but a definitive signature of a community whose structure and function have been refined by natural selection for a life dependent on fungal-derived nutrients.

In terms of the three evaluated fungal tissues, our findings demonstrate a significantly enriched predicted metabolic potential in the bacteriome associated with the fertile hymenophore of *L. asiatica*, compared to the stipe and pileus, which are sterile fungal tissues (Figure 10). The analysis of predictive functional pathways revealed a marked disparity in metabolic potential across the fungal tissues. The hymenophore-associated bacteriome demonstrated functional predominance, exhibiting the highest predicted activity in 19 of the 33 identified pathways (58%). This contrasted with the pileus and stipe tissues, where the resident bacteriomes showed predominance in only 10 (30%) and 2 (6%) pathways, respectively. The pronounced functional gradient, with a near ten-fold reduction in predominant pathways from the hymenophore to the stipe, underscores a significant stratification of metabolic roles within the fruiting body’s bacteriome. This distribution suggests that the bacterial community is not homogenously distributed but is highly structured, with its most robust and diverse metabolic contributions being localized to the fertile, spore-producing hymenophore. This finding strongly implies a tissue-specific functional specialization within the fungal holobiont, where the bacteriome’s role is most critical in supporting reproductive functions. This tissue-specific functional disparity suggests the hymenophore constitutes a distinct ecological microhabitat, fostering bacteria with heightened metabolic, cellular, and genetic processing activities. From a physiological perspective, the elevated functional capacity implies a critical symbiotic role for these bacteria in supporting the fungal reproductive effort. The enrichment of metabolic pathways suggests active bacterial involvement in nutrient acquisition, vitamin synthesis, or the provision of metabolic precursors essential for sporulation. Concurrently, the upregulation of cellular processes may indicate a role in biofilm formation or the biosynthesis of antimicrobial compounds, serving to protect the developing spores from pathogens and competitors. This represents a potential division of labor where the fungus outsources key physiological functions to its bacterial partners. Evolutionarily, the localization of this metabolically active bacteriome to the reproductive tissue indicates a co-evolved mutualism under strong selective pressure. Fungal genotypes that recruit and maintain such beneficial microbial communities would gain a direct fitness advantage through enhanced spore viability, protection, and dispersal success. This functional specialization suggests the hymenophore has evolved not merely as a fungal structure, but as a complex holobiont interface. Consequently, the reproductive success of *L. asiatica* appears intrinsically linked to the metabolic activity of its tissue-specific bacteriome, redefining the basidiomata as a finely integrated meta-organism, as previously pointed out in this contribution.

In terms of the comparative predictive functional pathways recorded in the four geographical provenances, the observed combination of high taxonomic richness and greater equitability in the endophytic bacterial community in basidiomata from Site 2, compared to Sites 1, 3, and 4, can be plausibly explained by the interplay between physiographic characteristics of the forest ecosystem in Site 2 and their associated metabolic resource diversity (Figure 11) The data indicate that bacteriome associated with basidiomata from Site 2 is distinguished by a significantly enhanced functional potential, particularly in core metabolic pathways such as vitamin and nucleotide metabolism. Of the 33 predictive functional pathways identified, the bacteriome associated with *L. asiatica* basidiomata exhibited the highest activity in 17 pathways (52%) at site 2. Site 3 displayed the highest activity in 10 pathways, while sites 1 and 4 showed substantially lower functional potential, with the highest activity in only 3 and 2 pathways, respectively. This enrichment in site 2 is likely a direct consequence of its location within a mature *Pinus yunnanensis-Quercus* mixed forest, which provides in turn a greater diversity of host-derived organic compounds. This elevated and diverse metabolic capacity creates a wider array of ecological niches. A community limited to a few core functions typically favors a small number of competitive generalist taxa, leading to lower evenness. In contrast, the broad functional repertoire at Site 2 facilitates niche partitioning, allowing for the coexistence of both dominant taxa of the bacteriome, which may harbor a larger number of rare taxa, that can specialize in the metabolism of more specific compounds (e.g., complex lipids or distinct polysaccharides). Consequently, the high total pathway abundance does not merely indicate a larger bacteriome community associated with basidiomata, but a more functionally diversified one. This diversification reduces competitive exclusion by providing distinct metabolic functions for different taxa to fulfill, thereby supporting both high species richness and a more equitable distribution of taxa by preventing any single group from achieving overwhelming dominance.

The heightened metabolic activity observed in the hymenophore-associated bacteriome suggests a direct and significant role in the basidiomata chemical ecology. Regarding bioactive compounds, this bacterial community is a potential source of novel metabolites. The enrichment of pathways in Metabolism and Genetic Information Processing indicates a high biosynthetic potential, which could lead to the direct production of antimicrobial, antifungal, or cytotoxic compounds by the bacteria themselves, serving to protect the valuable spore-producing tissue. Alternatively, through cross-kingdom signaling, the bacterial metabolome could induce or upregulate the fungal host’s own biosynthetic gene clusters, stimulating the production of fungal secondary metabolites with bioactive properties. Consequently, the different fungal tissues may constitute a previously underestimated reservoir of unique chemical diversity. Concerning organoleptic properties, including the taste, aroma, and color of *L. asiatica*, the tissue-specific bacteriome is a critical factor. The volatile organic compounds (VOCs) that define a mushroom’s aroma are often secondary metabolites or byproducts of primary metabolism. A metabolically hyperactive bacterial community in the studied fungal tissue compartments would be a prolific source of such VOCs, including sulfur compounds, pyrazines, and aldehydes. Similarly, bacterial processing of primary substrates (e.g., lipids and amino acids) can generate key flavor precursors, contributing to taste complexity. The stark contrast in metabolic potential between tissues implies that the sensory profile is not uniform but is a composite of distinct contributions from differently specialized microbiomes. Therefore, the culinary and sensory characteristics of the fruiting body are likely an emergent property of the fungal holobiont, shaped significantly by its structured bacterial consortia.

In summary, this study provides robust evidence to evaluate the proposed hypotheses. (i) Firstly, significant variation in bacteriome community structure and predictive functional pathways was confirmed across geographic provenances, irrespective of fungal tissue, thus supporting the initial hypothesis. (ii) Secondly, and contrary to the original prediction, the hymenophore exhibited significantly lower bacteriome richness and diversity than the pileus or stipe. This pattern suggests the fertile tissue is not a passive surface but a highly selective, functionally specialized niche, where physiological and evolutionary pressures favor a controlled, low-diversity consortium to ensure reproductive output; consequently, the second hypothesis is rejected. (iii) Finally, analytical evidence supports the third hypothesis, demonstrating that geographic provenance is the predominant driver of bacteriome variation. Ordination analyses revealed that community differentiation between locations surpassed that observed between anatomical tissues, establishing that geographical heterogeneity exerts a stronger influence on both the structure and functional potential of the *L. asiatica* bacteriome than intra-basidioma tissue differentiation.

To build upon these findings, future research should pursue an integrated, multidisciplinary framework along the following articulated avenues: (i) expanded biogeographic and ecological sampling to fully dissect the dominant effect of geographic origin (R^2^ = 0,46), sampling must be expanded across the entire 1500–3500 m elevation gradient inhabited by *L. asiatica* in southwestern China. This should explicitly encompass contrasting vegetation types to disentangle the effects of climate, host genotype, and plant community composition on the bacteriome. (ii) Quantitative abiotic modeling: the influence of specific edaphic and climate variables must be quantified using multivariate statistical models. Redundancy analysis (RDA) or Canonical Correspondence Analysis (CCA) should be employed to model the variance in bacteriome community composition explained by measured parameters, including soil pH, organic carbon content, total and available nitrogen, mean annual temperature, and precipitation. (iii) Functional validation of core taxa in vitro and *in planto*: the predicted functions of core genera (e.g., *Pantoea*, *Sphingomonas, Burkholderia*) require empirical validation. This involves the following: (a) These taxa must be isolated in pure culture. (b) Their capacity for saprotrophic competence must be quantified via assays for lignocellulolytic enzyme activity (e.g., cellulase on carboxymethylcellulose agar, laccase on ABTIS media). (c) Potential plant-growth-promoting traits such as nitrogen fixation must be confirmed, by using acetylene reduction assays, phosphate solubilization, by growing them in Pikovskaya’s medium, and phytohormone production (e.g., IAA via Salkowski’s reagent). (d) Gnobiotic systems must be developed to reinoculate axenic fungal cultures and confirm symbiotic functions *in planto.* (iv) Multi-omics must be integrated for dialogue elucidation. To move beyond correlation and infer causations, host–bacteria dialogue must be deciphered through dual RNA-seq (transcriptomics). This approach will simultaneously profile gene expression in both the fungal host and its bacteriome, elucidating how bacterial metabolites, for example, B-complex vitamins, auxins, siderophores modulate key fungal development pathways and metabolic genes, particularly those involved in basidiomata morphogenesis [12]. (v) Dissection of trophic networks via Stable Isotope Probing (SIP) is critical to empirically trace nutrient fluxes within the plant-fungus-bacteria continuum. Pulse-labeling of host plants with ^13^CO_2_ will allow tracing of photosynthate-derived carbon into fungal hyphae and subsequently into bacterial endophytes. Conversely, labeling with ^15^N_2_ gas can identify nitrogen fixing bacteria and trance the incorporation of fixed nitrogen into fungal tissues, thereby refining our understanding of the ecological niche occupied by *L. asiatica*.

## 5. Conclusions

Collectively, this study demonstrates a hierarchical model of community assembly for the endophytic bacteriome within *L. asiatica* basidiomata. The primary structuring force is a broad-scale environmental filter (geography and climate), which establishes the regional species pool, while secondary tissue-specific microhabitat selection acts as a fine filter, shaping the final community structure within individual sporomes. The identified core bacterial taxa display evolutionary signatures of co-adaptation with some congeneric Boletales, yet local habitat heterogeneity and functional demands imprint distinct community identities. These findings bridge a critical conceptual gap in the ecology of *Lanmoa* and provide a robust exemplary model for understanding multipartite symbioses within the ecologically crucial agaricomycetes. Beyond theoretical advances, this work furnishes a practical mechanistic framework for applications in geographic provenance tracing (e.g., authenticating the origin of wild-foraged mushrooms) and informs strategies for the conservation of this prized edible ectomycorrhizal mushroom by defining its core beneficial microbiome. This establishes a foundational platform for future investigations aimed at harnessing fungal microbiomes for biotechnological exploitation and conservation.

## Figures and Tables

**Figure 1 microorganisms-13-02431-f001:**
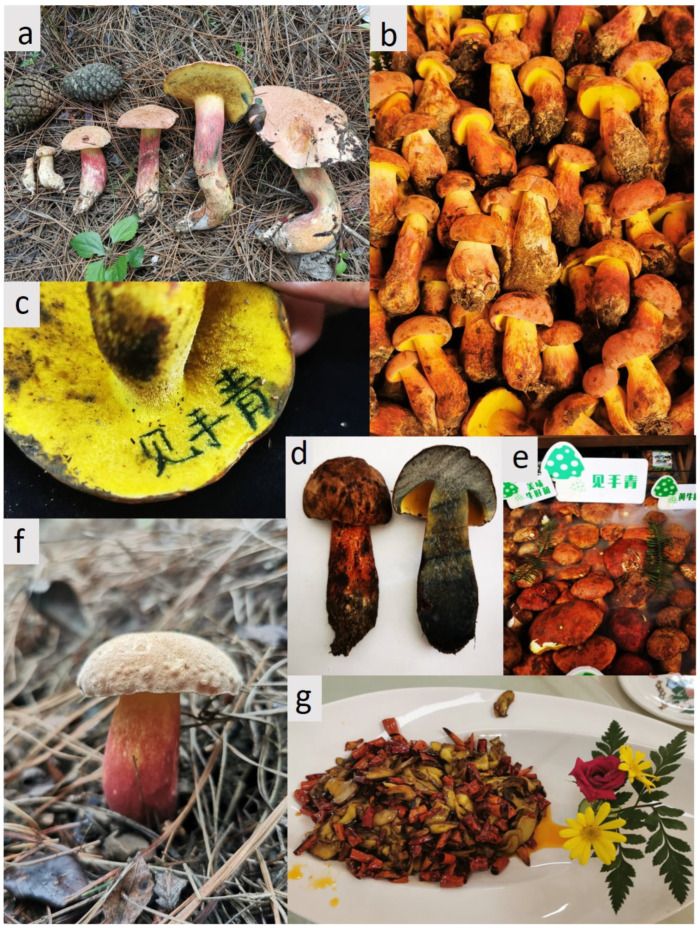
Morphological characteristics and sociocultural context of *Lanmoa asiatica*. (**a**) Developmental stages of basidiomata in a *Pinus yunnanensis* forest in Yunnan, China. (**b**) Commercial-scale vending of fresh basidiomata at the Mushuijua market, Kunming, China. (**c**) Blue bruising reaction (cyanescence) on the hymenophore, showing the Chinese characters for its common name, “Jiàn Shǒu Qīng” meaning literally “see-hand-blue/green”. (**d**) Cyanescent reaction present on the pileus and stipe contexts. (**e**) Fresh basidiomata of Lanmoa asiatica displayed for sale at a restaurant, with a label indicating its common name, 见手青 (Jiàn Shǒu Qīng). The name translates literally as “see-hand-blue/green,” referring to its characteristic blueing reaction (cyanescence) upon handling. (**f**) Young basidioma growing within in a mixed *Pinus yunnanensis*-*Quercus* spp. forest, its natural habitat near Kunming, Yunnan Province, China. (**g**) A typical preparation of fried *L. asiatica* with hot chili, a culinarily valued dish in southwestern China.

**Figure 2 microorganisms-13-02431-f002:**
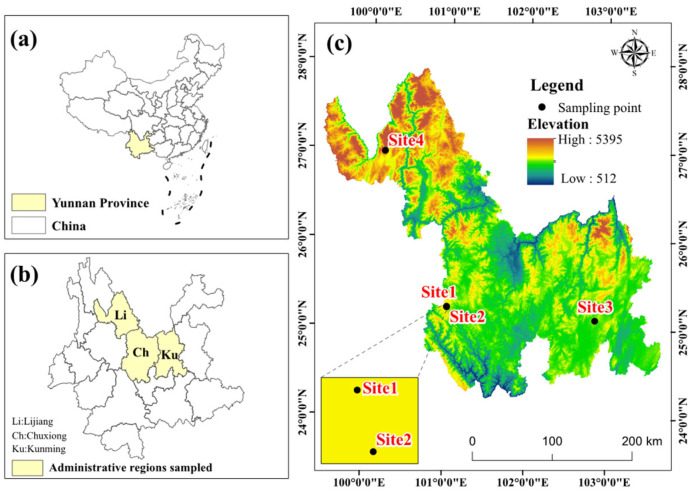
Geographic distribution of sampled *Lanmoa asiatiaca* provenances within Yunnan Province, China. (**a**) Map of China indicating the location of Yunnan Province. (**b**) The three counties in Yunnan from which basidiomata were collected. (**c**) Spatial distribution of the four collection sites (denoted as points 1–4), overlaid on a digital elevation model. An inset provides greater cartographic detail for Sites 1 and 2 to resolve their relative proximity. Site designations correspond with those detailed in Table 1.

**Figure 3 microorganisms-13-02431-f003:**
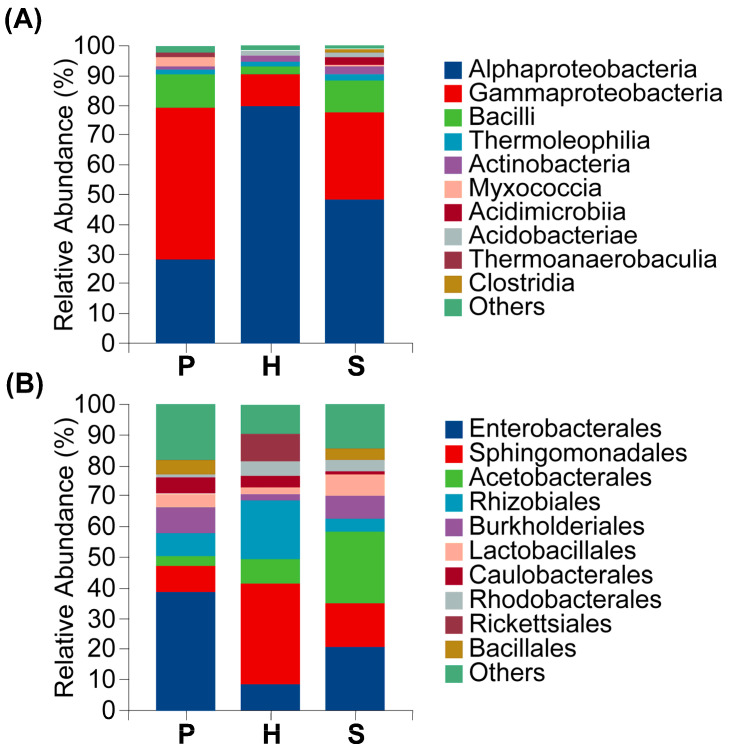
Relative abundance of the ten most abundant bacterial classes (**A**) and orders (**B**) inhabiting different anatomical compartments of *Lanmaoa asiatica* basidiomata. The “Others” category aggregates OTUs below the detection threshold and unclassified lineages. Values are presented as the mean of three biological replicates (*n* = 3). Compartments are denoted as follows: P, pileus; H, hymenophore; S, stipe.

**Figure 4 microorganisms-13-02431-f004:**
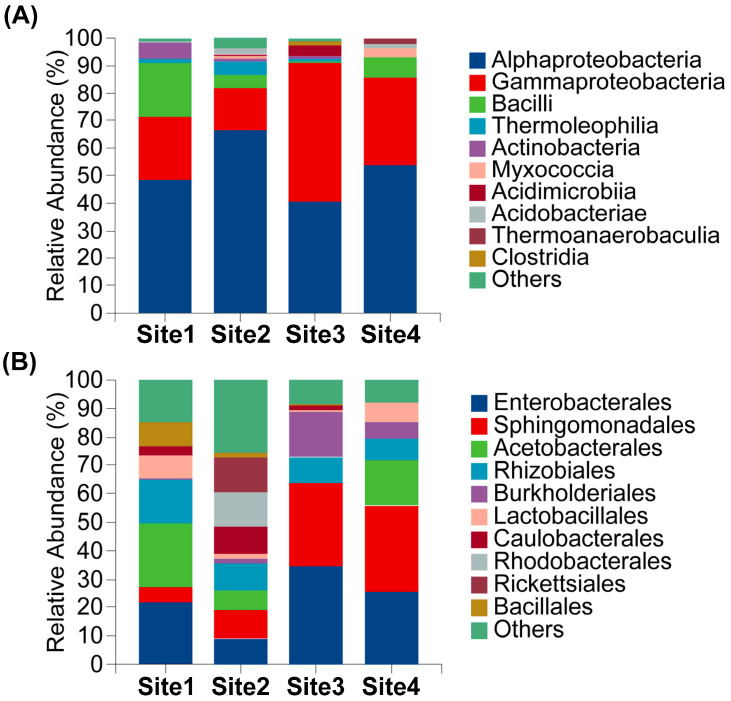
Relative abundance of the ten most abundant classes (**A**) and orders (**B**) of endophytic bacteria within *Lanmaoa asiatica* basidiomata collected from four distinct geographic provenances. Taxa below the detection threshold and unclassified lineages are consolidated into “Others.” Bars represent the mean values of three biological replicates. Corresponding site descriptions for provenance codes Site 1 through Site 4 are provided in Table 1.

**Figure 5 microorganisms-13-02431-f005:**
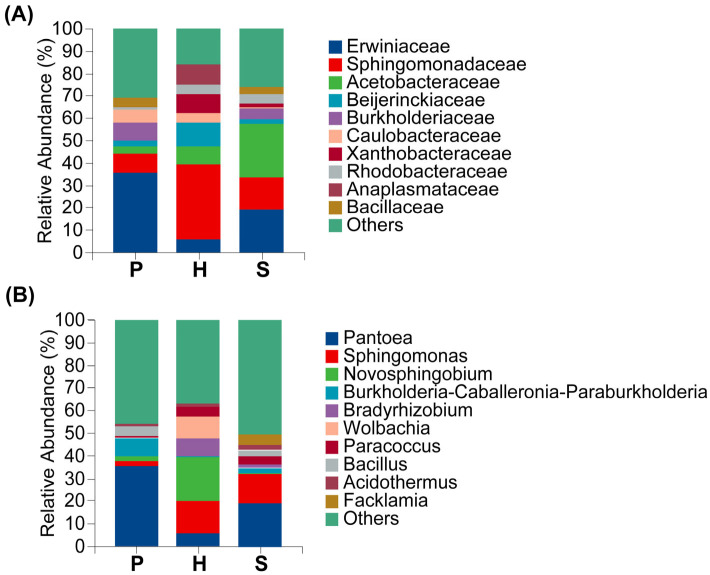
Relative abundance of the ten most abundant bacterial families (**A**) and genera (**B**) inhabiting different anatomical compartments of *Lanmaoa asiatica* basidiomata. The “Others” category aggregates OTUs below the detection threshold and unclassified lineages. Values are presented as the mean of three biological replicates (*n* = 3). Compartments are denoted as follows: P, pileus; H, hymenophore; S, stipe.

**Figure 6 microorganisms-13-02431-f006:**
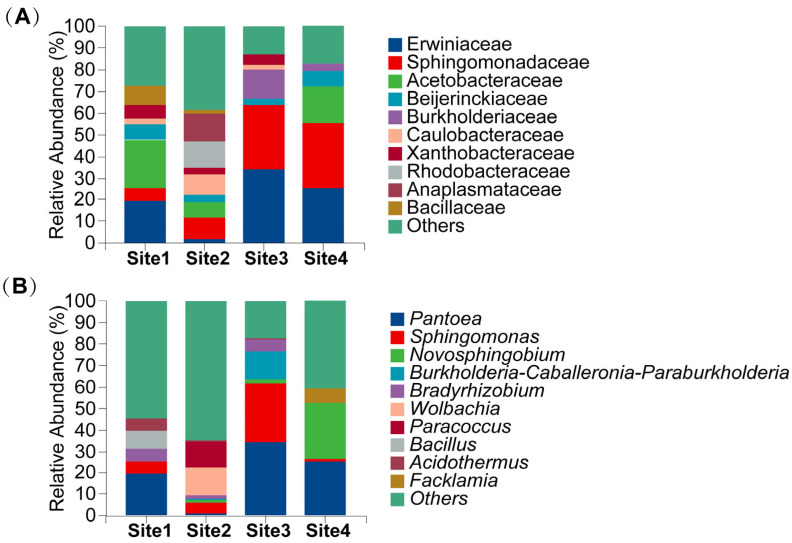
Relative abundance of the ten most abundant families (**A**) and genera (**B**) of endophytic bacteria within *Lanmaoa asiatica* basidiomata collected from four distinct geographic provenances. Taxa below the detection threshold and unclassified lineages are consolidated into “Others.” Bars represent the mean values of three biological replicates. Corresponding site descriptions for provenance codes Site 1 through Site 4 are provided in Table 1.

**Figure 7 microorganisms-13-02431-f007:**
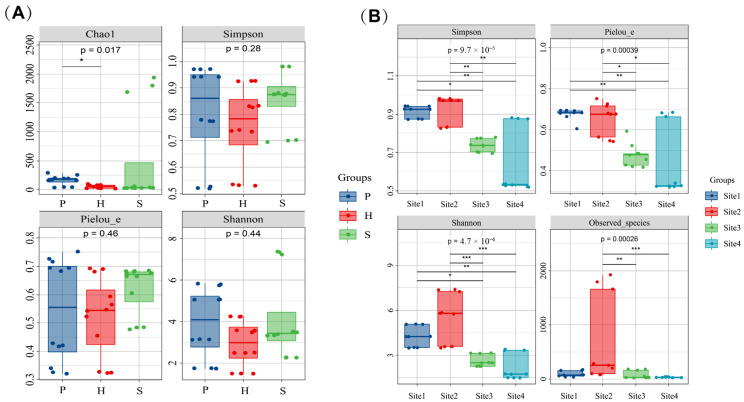
Alpha diversity of endophytic bacteria within *Lanmoa asiatica* basidiomata across anatomical sections (**A**) and geographical locations (**B**). Boxplots depict variations in diversity indices. Horizontal lines above the boxplots denotate statistically significant pairwise comparisons derived from *t*-tests; asterisks indicate significance levels (* *p* < 0.05, ** *p* < 0.01, *** *p* < 0.001). Dark horizontal lines connect groups exhibiting significant differences in mean values. Abbreviations for basidiomata tissues (**left**) and geographic sampling locations (**right**) correspond to those specified in Figure 3 and Table 1, respectively.

**Figure 8 microorganisms-13-02431-f008:**
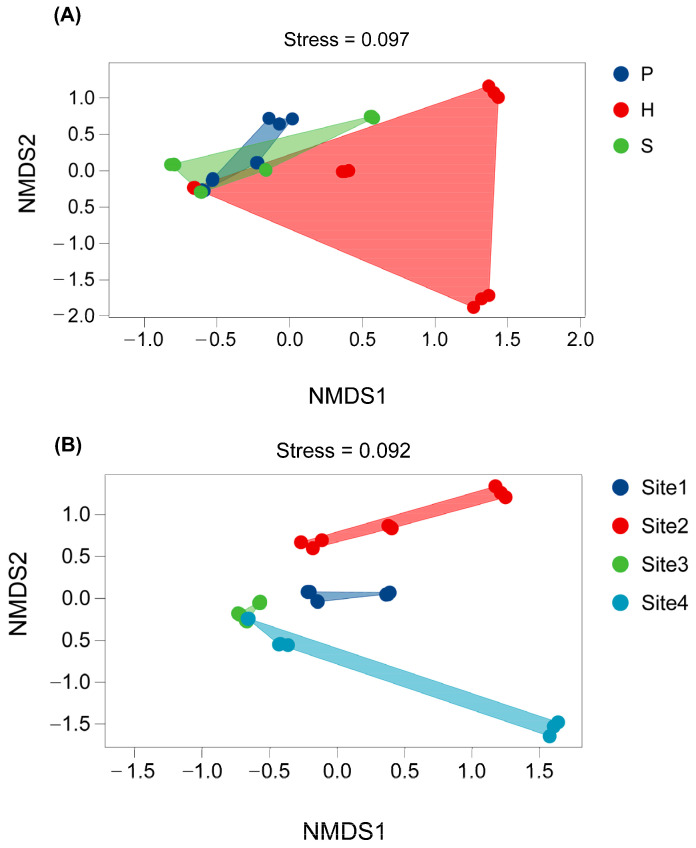
Variation in the endophytic bacterial community structure of *Lanmoa asiatica* driven by (**A**) anatomical section and (**B**) geographical location. Non-metric multidimensional scaling (NMDS) ordinations visualize community composition based on Bray–Curtis dissimilarity matrices.

**Figure 9 microorganisms-13-02431-f009:**
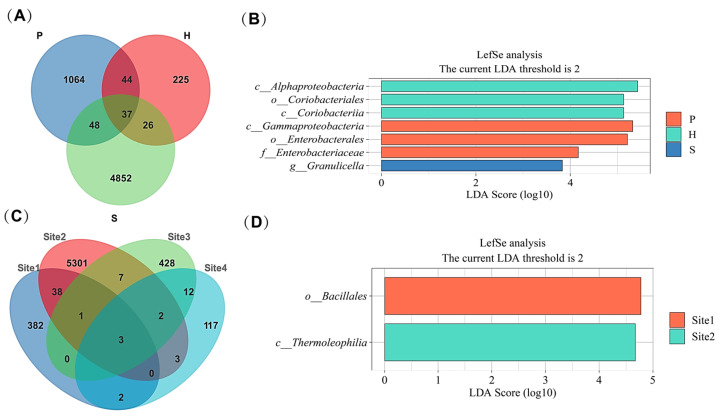
Core and differential microbiota within *Lanmoa asiatica* basidiomata. (**A**,**C**) Petal diagrams depicting the distribution of shared (core) and unique bacterial Amplicon Sequence Variants (ASVs) across distinct anatomical parts (**A**) and geographical locations (**C**). Petals represent sample groups, with numerical values indicating ASV counts. (**B**,**D**) Linear Discriminant Analysis Effect Size (LefSe) results identifying taxa differentially abundant across anatomical sections (**B**) and geographical locations (**D**). Histograms represent the Linear Discriminant Analysis (LDA) effect size (log base 10 scale) for discriminatory taxa. Abbreviations of anatomical parts and geographical locations are those defined in Figure 3 and Table 1, respectively.

**Figure 10 microorganisms-13-02431-f010:**
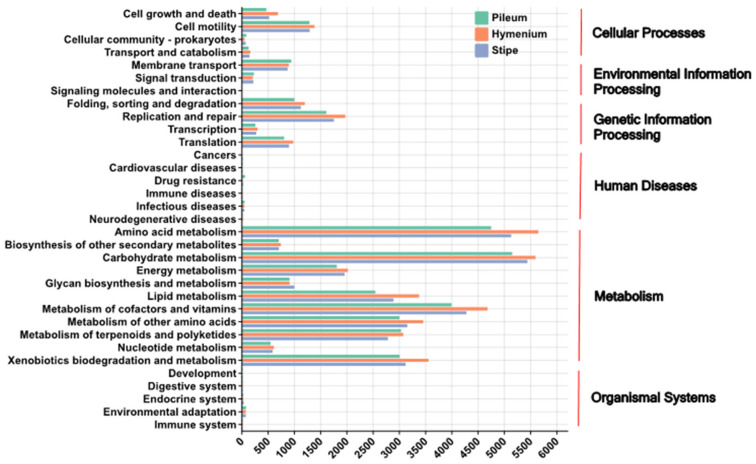
Compartment-specific differential abundance of predicted bacterial metabolic pathways in *Lanmaoa asiatica* basidiomata, based on PICRUSt2 and KEGG analysis.

**Figure 11 microorganisms-13-02431-f011:**
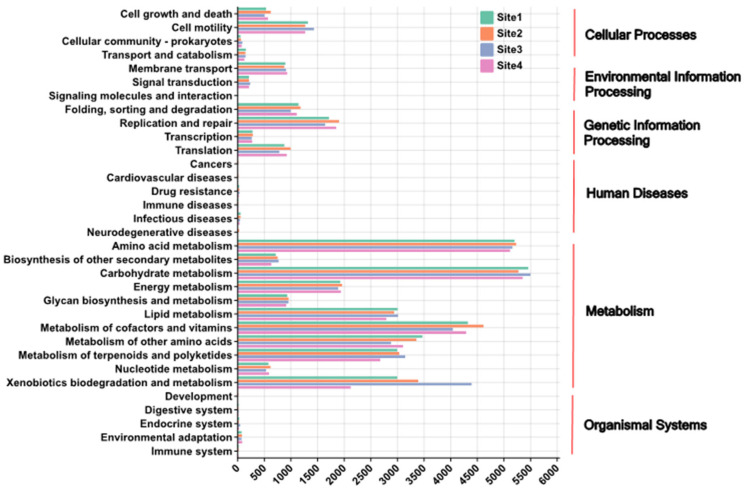
Geographical provenance-specific differential abundance of predicted bacterial metabolic pathways in *Lanmaoa asiatica* basidiomata, based on PICRUSt2 and KEGG analysis. Corresponding site descriptions for provenance codes Site 1 through Site 4 are provided in Table 1.

**Table 1 microorganisms-13-02431-t001:** General geographical and vegetative characteristics of the investigated sites.

Site Number	Location	Altitude (m)	Longitude (E)	Latitude (N)
S1	ChuXiong perfecture, NanHua county, WuJie town	2583	101°0′11″	25°14′40″
S2	ChuXiong perfecture, NanHua county, WuJie town	2583	101°0′19″	25°14′15″
S3	Shuanglong Sub-district, Panlong District, Kunming City	2000	102°50′49″	25°6′49″
S4	LiJiang city, YuLong county, Baisha town	3147	100°10′30″	26°58′36″

## Data Availability

The original contributions presented in this study are included in the article/Appendix A. Further inquiries can be directed to the corresponding authors.

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
