# Peer review of "Geographic Provenances Outweigh Tissue Compartments in Bacteriome Assembly of the Ectomycorrhizal, Edible, and Hallucinogenic if Undercooked, *Lanmoa asiatica* (Boletaceae, Boletales) Mushroom from Yunnan China"

_microorganisms, 2025, doi:10.3390/microorganisms13112431_

Round 1
Reviewer 1 Report
Comments and Suggestions for Authors
Why did site 2 possess not only high richness but also a more equitable distribution of dominant and rare taxa?
Is this why site 2 consistently exhibits the highest functional potential, evidenced by significantly elevated total metabolic pathway abundance?
How could it affect the production of bioactive compounds or organoleptic properties of fruiting bodies?
Why is there a predominance of gram-negative bacteria, which occurred in this coevolutionary process, where gram-positive bacteria are low in abundance?
Possible answers to these questions could be incorporated in the discussion.
Author Response
ANSWER TO REVIEWERS
Journal: Microorganisms
Reference number: microorganisms-3907434
Authores: Man Guo, Dong Liu, Zhilan Xia, Tao Xie, Luofeng Su, Jesus Pérez-Moreno, Fuqiang Yu
Manuscript title: “Geographic provenances outweigh tissue compartments in bacteriome assembly of the ectomycorrhizal, edible, and hallucinogenic if undercooked, Lanmoa asiatica (Boletaceae, Boletales) mushroom from Yunnan China”
Assigned Editor: Diane Dai
We extend our sincere gratitude to four reviewers for their insightful comments, valuable suggestions, necessary corrections, and thorough review of the manuscript. All feedback has been carefully considered and addressed in the revised version. To facilitate the review process, the following documents are provided: a) A point-by-point response table addressing each of the reviewers’ suggestions; and b) An improved version of the manuscript, in which all modifications have been highlighted in yellow for ease of identification. This revised version has been substantially enhanced through the incorporation of the reviewers’ generous recommendations. New paragraphs have been written, particularly to enrich the Discussion section. Thank you once again for the constructive and detailed input.

Reviewer 2 Report
Comments and Suggestions for Authors
Dear,
The paper "Geographic provenances outweigh tissue compartments in bacteriome assembly of the ectomycorrhizal, edible, and hallucinogenic if undercooked, Lanmoa asiatica (Boletaceae, Boletales) mushroom from Yunnan, China" is well-written and provides relevant information on the interaction between bacteria and Basidiomycota. Some suggestions are listed below:
- Review the graphic quality of the maps;
- Names of orders, families, genera, and species should be written in italics. Review the entire document;
- In Figure 8, the part of the fungus considered should be specified, as well as the collection sites. As presented, it appears that all data were grouped, without considering the factors studied (parts of the basidiomycete and collection sites);
- Lines 568-574: review the statements, as these soil parameters were not evaluated;
- It is important that each hypothesis postulated in the introduction be clearly revisited in the discussion, indicating whether it was corroborated or refuted.
After these adjustments, the article can be considered for publication.
Sincerely,
Author Response

(The authors gave the same response as above.)

Reviewer 3 Report
Comments and Suggestions for Authors The manuscript is a professionally well-developed, thorough and demanding work.Its language is easy to understand and its message is easy to follow.
The body of the text is well and neatly edited. Suggested minor corrections: Lines 90 – 92: missing source Lines 182 and 191: operating is not necessary, it is not a good term. "working" would be more appropriate. Line 193: it is not clear whether the extraction lysis buffer comes from the kit used. Line 227: family taxonomic level, abundance taxon instead of abundant family Line 281: taxa instead of families Lines 292 and 294: genus and genera instead of taxa and taxon Ad 1007, 1099, 1124 Please write the scientific species and genus names in italics. Ad 1010 correctly: Streptomyces sp. ad l. 1077-78: citation to be clarified ad l. 1127 correctly: sp. ad l. 1139 correctly: sp
Author Response

(The authors gave the same response as above.)

Reviewer 4 Report
Comments and Suggestions for Authors
The submitted manuscript entitled “Geographic provenances outweigh tissue compartments in bacteriome assembly of the ectomycorrhizal, edible, and hallucinogenic if undercooked, Lanmoa asiatica (Boletaceae, Boletales) mushroom from Yunnan China” presents a comprehensive and multifaceted investigation into the composition and functional attributes of the bacteriome associated with the ectomycorrhizal mushroom Lanmoa asiatica. By examining specimens from four distinct geographic origins and partitioning each into three tissue types, the authors employ advanced genetic and bioinformatic analyses to elucidate endophytic bacterial diversity, community structure, and metabolic potential. The findings reveal that geographic provenance has a stronger structuring effect than intra-tissue differentiation, providing an important conceptual contribution to our understanding of fungal microbiome assembly. The work is extensive, rich in data, and grounded in a robust theoretical framework, offering both fundamental insights and practical applications in provenance authentication and conservation strategies for this valued edible mushroom. The manuscript is well-prepared, the discussion is thorough and precise, the references are relevant to the topic. While the work is of good quality, it would benefit from minor revisions, particularly in terms of editorial clarity, to ensure its findings are fully accessible and impactful for the research community.
Details below:
Comment 1#
Line 124: Did storing the freshly collected fruiting bodies in plastic bags not affect the condition of the bacteriome? In my opinion, sterile glass or plastic containers would have been more appropriate. Plastic bags, even if not tightly sealed but merely rolled, could potentially have influenced the condition of the fruiting bodies.
Comment 2#
Figure 1 The axis labels on the figures, as well as the figures themselves (maps), are illegible. The maps are missing the markings (a, b, c) that are included in the caption. Please improve the quality of the figures.
Comment 3#
Line 135: According to the journal's guidelines, the table caption should be placed above the table.
Comment 4#
I have a doubt whether using only 3 fruiting bodies from a given site constituted a representative sample for the study.
Comment 5#
Figure 3 Improve the quality of the presented results. Add spaces after the caption below the figure (Line 276). Adjust the caption format according to the journal's requirements.
Comment 6#
Lines 285, 298, 303, 312, 362, 374. Correct the spacing.
Comment 7#
Lines 336, 356, 401, 405: The terms “Supplementary Figure 1A” (Supplementary Figure 1B), Supplementary Table S1A), and Supplementary Table S1B) are missing the word "Supplementary" in the submission. Please add "Supplementary materials" where they are missing.
Comment 8#
Line 338: (: what does that mean?
Comment 9#
Line 368: additional dot
Comment 10#
Figure 6: Improve the quality of the results presented.
Comment 11#
Figure 8: Improve the quality of captions in the figure.
Comment 12#
Line 513: correct italically
Author Response

(The authors gave the same response as above.)
